materials science

peanut meal-based wood adhesive, urea, epichlorohydrin, properties, water resistance

**Authors for correspondence:**
Fusheng Chen
e-mail: fushengc@haut.edu.cn
Boye Liu
e-mail: liuboye@aliyun.com

This article has been edited by the Royal Society of Chemistry, including the commissioning, peer review process and editorial aspects up to the point of acceptance.

# Peanut meal-based wood adhesives enhanced by urea and epichlorohydrin

Chen Chen, Fusheng Chen, Boye Liu, Yan Du, Chen Liu, Ying Xin and Kunlun Liu

College of Food Science and Technology, Henan University of Technology, Zhengzhou, Henan Province 450001, People's Republic of China

CC, 0000-0001-5914-228X; FC, 0000-0002-8201-1234; BL, 0000-0003-4084-0357

Peanut meal (PM) has recently emerged as a potential protein source for wood adhesives, owing to superior features such as high availability, renewability and eco-friendliness. However, the poor properties of unmodified PM-based wood adhesives, compared with their petroleum-derived counterparts, limit their use in high-performance applications. In order to promote the application of PM-based wood adhesives in plywood industry, urea (U) and epichlorohydrin (ECH) were used to enhance the properties of the adhesives and the modification mechanism was investigated. PM-based wood adhesives made with U and ECH were shown to possess sufficient water resistance and exhibited higher apparent viscosity and solid content than without. Fourier-transform infrared spectroscopy results suggested that U denatured PM protein and expose more reactive groups, allowing ECH to react better with U-treated PM protein to form a dense, cross-linked network which was the main reason for the improvement of the properties. The crystallinity increased from 2.7% to 11% compared with the control, indicating that the molecular structure of the resultant adhesive modified by U and ECH became more regular and compact owing to the cross-linked network structure. Thermogravimetry tests showed that decomposition temperature of the protein skeleton structure increased from 307°C to 314°C after U and ECH modification. Scanning electron microscopy images revealed that using U and ECH for adhesives resulted in a smooth protein surface which prevented moisture penetration and improved water resistance. PM-based adhesives thus represent potential candidates to replace petroleum-derived adhesives in the plywood industry, which will effectively promote the rapid development of eco-friendly adhesives and increase the added value of PM.

# 1. Introduction

With the rapid development of the artificial plywood industry, the production and use of wood adhesives have steadily increased in recent years. At present, the vast majority of the adhesives used in plywood composites are based on formaldehyde. Typical examples are urea (U)-, phenol- and melamine-formaldehyde resins. Because of their excellent bonding properties, high water resistance and low cost, these are very important in the wood adhesive industry [1,2]. However, the release of formaldehyde (derived from non-renewable petroleum resources) during the production and use of these resins is a crucial problem that needs to be addressed. The depletion of petroleum resources and the increasing awareness of environmental protection issues [3,4] have created demand for adhesives that are free of formaldehyde and made from renewable bioresources [5].

Plant proteins are high-quality renewable resources for formaldehyde-free wood adhesives, and have received widespread attention. A number of previous studies have focused on plant protein-based adhesives, including adhesives based on soy [6], cotton [7], wheat [8] and sesame proteins [9]. Nevertheless, there are few studies on the preparation of wood adhesives based on peanut proteins. China is the world's principal peanut producer and consumer, with a total output of about 17.40 million tons in 2017 [10]. These abundant resources provide sufficient raw materials for the study and development of peanut meal (PM) protein and processed products. As a main by-product of peanut oil processing, PM is primarily used as animal feed or crop fertilizer owing to its high protein content, which results in severe waste of resources. Therefore, the efficient use of PM represents an issue of significant interest. Three major components, arachin, conarachin I and conarachin II, containing a large number of reactive groups, account for greater than 75% of the total protein content in peanut. Chemical reactions can take place between these reactive groups and modifiers, which makes it possible to prepare wood adhesives from PM [11].

The poor properties of unmodified plant protein-based wood adhesives limit their use in high-performance applications. Methods that can modify plant protein-based adhesives to enhance their properties include denaturation [12] or modifications with cross-linking agents [13], synthetic resins [14], nanomaterials [15,16] and biomimetic materials [17,18]. These modifications have produced effective improvements in the properties of plant protein-based adhesives [19].

Considering the high content of reactive groups in peanut protein, chemical modifications were selected to enhance the properties of PM-based adhesives. Epichlorohydrin (ECH) can react with the reactive groups of the proteins via its epoxy group and generate new firm chemical bonds [20], which make the interior of the protein structure more stable and less susceptible to damage from the external environment. However, these reactive groups are generally embedded inside the protein and cannot easily interact with the ECH, which makes it difficult to meet the needs of ECH modification alone. Therefore, it is necessary to add a denaturing agent before adding ECH in order to get a better reaction. U is used as a denaturing agent to break the structure of the protein and expose the functional groups [21], so that they can react with ECH sufficiently, thus further improving the properties of PM-based adhesives.

Accordingly, in order to promote application of PM in the wood adhesives industry and the efficient use of PM, U and ECH were employed to enhance the properties of PM-based adhesives. The dry and wet shear strength, viscosity and solid content were determined. The corresponding modification mechanism was characterized by various analytical methods.

# 2. Experimental section

## 2.1. Materials

Defatted PM (50.12% protein content, 80 meshes) with moisture and crude fat contents of 8.19% and 1.81%, respectively, was purchased from Zhengyang Sannong Seed Industry Co., Ltd (Henan, China). U and ECH (analytical grade) were purchased from Luoyang Reagent Co., Ltd (Henan). Poplar veneer ($300 \times 300 \times 2.0$ mm, 9%–11% moisture content) was obtained from Bio Biologic Co., Ltd (Henan).

## 2.2. Preparation of peanut meal-based adhesives

To prepare adhesive A, 20 g dried defatted PM was mixed with 80 ml pure water and stirred in a three-necked flask equipped with a condenser (50°C, 120 min, 300 r min$^{-1}$). For adhesive B, 20 g dried defatted PM was mixed with 80 ml 1 mol l$^{-1}$ U solution and stirred in a three-necked flask equipped with a

condenser (50°C, 120 min, 300 r min$^{-1}$). To prepare adhesive C, 20 g dried defatted PM was mixed with 80 ml 1 mol l$^{-1}$ U solution and stirred in a three-necked flask equipped with a condenser (50°C, 60 min, 300 r min$^{-1}$), then ECH was added (30%, based on the mass of PM) and the mixture was stirred again (50°C, 60 min, 300 r min$^{-1}$). For adhesive D, 20 g dried defatted PM was mixed with 80 ml pure water and stirred in a three-necked flask equipped with a condenser (50°C, 60 min, 300 r min$^{-1}$), then ECH was added (30%, based on the mass of PM) and the mixture was stirred again (50°C, 60 min, 300 r min$^{-1}$).

## 2.3. Dry and wet shear strength measurements

Chinese National Standard GB/T 9846-2015 was applied to measure shear strength of plywood (electronic supplementary material, preparation of three-ply plywood) using a MWD-10 testing machine (Si Da Te Test Co., Ltd., Shandong, China) at 10 mm min$^{-1}$. The dry shear strength of 12 test-pieces (100 × 2.5 mm) was tested without soaking treatment. The wet shear strength of 12 test pieces (100 × 2.5 mm) was measured after submerging in water (63 ± 2°C, 3 h). The force required to break the glued wood specimen was recorded, and the bond strength was calculated from equation (2.1) ($n = 12$):

$$\text{bond strength (MPa)} = \frac{\text{force (N)}}{\text{gluing area (m}^2)}. \tag{2.1}$$

## 2.4. Apparent viscosity measurements

An RS-6000 HAAKE rotational rheometer (Thermo Fisher Scientific Corporation, Waltham, MA) was used to determine apparent viscosity of modified adhesives. About 1 ml of the adhesives was evenly distributed on the test plate. The distance between the rotor (P35) and the plate was 1 mm. Measurements were made at 25°C with a steady shear flow and shear rate 1–100 s$^{-1}$. The initial viscosity was recorded at a shear rate of 1 s$^{-1}$. In order to prevent the loss of moisture during the test, a small amount of silicone oil was evenly applied around the test plate.

## 2.5. Solid content measurements

Modified adhesive (approx. 3 g; weight $\alpha$) was put in an aluminium box and dried to constant weight at 105°C (weight $\beta$). Solid content was calculated from equation (2.2) ($n = 3$):

$$\text{solid content (\%)} = \frac{\beta}{\alpha} \times 100\%. \tag{2.2}$$

## 2.6. Fourier-transform infrared spectroscopy tests

The modified adhesives were dried to constant weight at 120 ± 2°C, then ground into powder that could pass through a 200-mesh sieve. Each sample was mixed with KBr (powder; 1 : 100 (w/w)) and pressed into transparent thin sheets for Fourier-transform infrared (FTIR) spectroscopy tests. The FTIR spectra of the modified adhesives were recorded from 400 to 4000 cm$^{-1}$ using a Nicolet 6700 spectrometer (Thermo Nicolet Corporation, Madison, WI; 4 cm$^{-1}$ resolution, 32 scans).

## 2.7. X-ray diffraction tests

The modified adhesives were dried to constant weight at 120 ± 2°C. X-ray diffraction (XRD) patterns were measured using a D8 ADVANCE diffractometer (Bruker, Germany) with a cobalt source and $2\theta$ values 5–80° at 40 kV and 40 mA.

## 2.8. Thermogravimetry tests

The modified adhesives were dried to constant weight at 120 ± 2°C, then ground into powder. Thermal stability was analysed using a TA Q50 instrument (Waters Company, USA). Approximately 10 mg of modified adhesive was scanned at 25 to 600°C (heating rate 10°C min$^{-1}$) under a nitrogen gas flow of 20 ml min$^{-1}$.

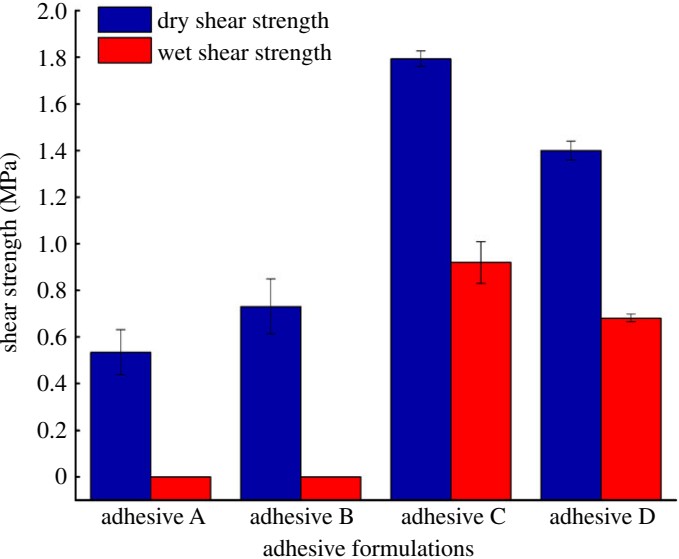

**Figure 1.** Dry and wet shear strength of the different adhesives (adhesive A: unmodified PM-based adhesive, adhesive B: PM-based adhesive modified by U, adhesive C: PM-based adhesive modified by U and ECH, adhesive D: PM-based adhesive modified by ECH).

## 2.9. Scanning electron microscopy tests

The surface structure of the modified adhesives was tested using a Quanta Feg 250 instrument (FEI Company, USA). The modified adhesives were dried to constant weight at $120 \pm 2°C$. Each sample was sputter-coated with gold before scanning electron microscopy (SEM).

# 3. Results and discussion

## 3.1. Dry and wet shear strength of the adhesives

Figure 1 shows dry and wet shear strengths of the modified adhesives. The dry shear strength of adhesive A was 0.53 MPa. The dry shear strength of adhesive B increased to 0.73 MPa. This might be owing to U expanding the globular structure of the protein, exposing the inner hydrophobic groups and increasing the contact area with the bonding surface. Treatment with U and ECH resulted in a 237.73% increase in the dry shear strength of adhesive C relative to that of adhesive A (from 0.53 to 1.79 MPa). The dry shear strength of adhesive D increased by 164.15% (from 0.53 to 1.40 MPa), relative to that of adhesive A.

The wet shear strength of adhesive A was 0 MPa, indicating that the water resistance of adhesive A prepared directly from PM was poor. The wet shear strength of adhesive B was the same as that of adhesive A, indicating that adding U only did not improve the water resistance. The wet shear strength of adhesive C was 0.92 MPa, sufficient to meet the requirements of Chinese National Standard GB/T 9846-2015 (greater than or equal to 0.70 MPa). We suggest that ECH reacted with functional groups of the PM protein unfolded by U to form a dense, three-dimensional network structure that resisted water intrusion. In addition, the denser network structure resulted in a stronger cohesion, which enhanced the water resistance; this is consistent with the results of Gui *et al.* [14]. For adhesive D, the wet shear strength was 0.68 MPa, higher than those of adhesives A and B but below the requirements of the national standard. This could be understood as follows: although ECH reacted with PM protein, without U treatment, the globular structure of the protein was not fully unfolded and its internal groups were not completely exposed, resulting in a less effective cross-linking reaction compared with that for adhesive C; therefore, the wet shear strength of adhesive D was lower than that of adhesive C, but its water resistance was still higher than those of adhesives A and B.

## 3.2. Apparent viscosity of the adhesives

Figure 2 shows apparent viscosities of the various adhesives, and table 1 shows the corresponding initial viscosities. The apparent viscosities of all the adhesives decreased when the shear rate increased from

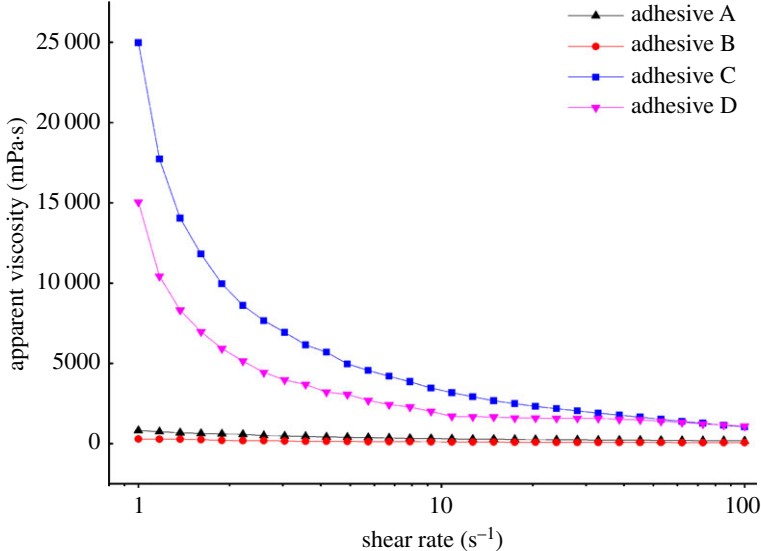

**Figure 2.** Apparent viscosity of the different adhesives (adhesive A: unmodified PM-based adhesive, adhesive B: PM-based adhesive modified by U, adhesive C: PM-based adhesive modified by U and ECH, adhesive D: PM-based adhesive modified by ECH).

**Table 1.** Initial viscosity of the different adhesives (adhesive A: unmodified PM-based adhesive, adhesive B: PM-based adhesive modified by U, adhesive C: PM-based adhesive modified by U and ECH, adhesive D: PM-based adhesive modified by ECH).

| adhesive formulation | adhesive A | adhesive B | adhesive C | adhesive D |
|---|---|---|---|---|
| initial viscosity (mPa·s) | 819.05 | 290.15 | 24 968 | 15 049 |

1 to 100 s$^{-1}$, revealing that they behaved as shear-thinning fluids. The initial viscosity of adhesive B was 290.15 mPa·s, compared with 819.05 mPa·s for adhesive A. The main reason for this decrease was that U unfolded the globular structure of the PM protein [22]. The viscosity of adhesives C and D, formed by adding ECH, increased to 24 968 and 15 049 mPa·s, respectively. The viscosity of adhesive C was higher than that of adhesives A and D. This might be owing to two main reasons: first, the treatment with U resulted in an unfolded protein structure exposing more reactive groups. ECH could then cross-link these reactive groups to form a network structure. The cohesion needed to maintain this network structure was mainly produced by chemical bonds, which made the protein structure denser and more viscous. Second, although ECH could still form cross-links with the PM protein, the unfolding of the protein structure and the exposure of reactive groups in adhesive D were less extensive than in adhesive C. The apparent viscosities exhibited essentially the same trend as observed for shear strength.

## 3.3. Solid content of the adhesives

Solid content is an important physical property of wood adhesives; typically, lower solid content means that more moisture must be removed from the adhesive, which may reduce the bonding performance during hot pressing processes [23]. However, excessive solid content results in an uneven distribution of adhesive and thus makes the coating process difficult. The solid content of the different adhesives in this study is shown in figure 3. The value measured for adhesive A was 21.60%. After adding U, the solid content of adhesive B increased to 22.31%, probably because U could unfold the globular structure of the PM protein, causing its rearrangement. Upon adding U and then ECH to PM, the solid content of adhesive C reached 25.14%. The solid content of adhesive D was 24.59%. The higher solid contents of adhesives C and D may be owing to a reaction between the epoxy groups of ECH and functional groups of PM protein, which caused the epoxy groups to graft onto the protein skeleton, resulting in an increase in solid content. Furthermore, the treatment with U resulted in a more unfolded protein structure in adhesive C, with higher exposure of reaction sites, leading to a more complete reaction with ECH and a higher solid content compared with adhesive D. The solid content showed essentially the same trend as apparent viscosity and shear strength.

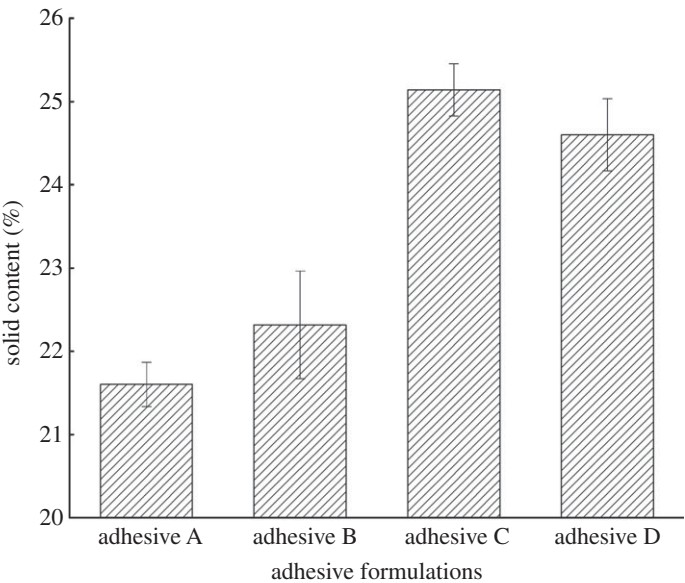

**Figure 3.** Solid content of the different adhesives (adhesive A: unmodified PM-based adhesive, adhesive B: PM-based adhesive modified by U, adhesive C: PM-based adhesive modified by U and ECH, adhesive D: PM-based adhesive modified by ECH).

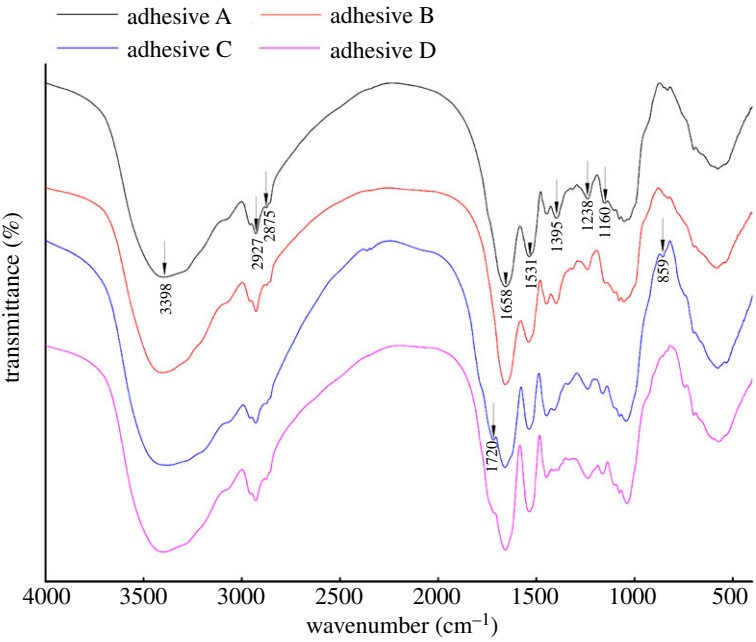

**Figure 4.** FTIR spectra of the different adhesives (adhesive A: unmodified PM-based adhesive, adhesive B: PM-based adhesive modified by U, adhesive C: PM-based adhesive modified by U and ECH, adhesive D: PM-based adhesive modified by ECH).

## 3.4. Fourier-transform infrared analysis

Figure 4 shows FTIR spectra of the different modified adhesives. The broad band observed around 3398 cm$^{-1}$ arose from stretching vibrations of O–H bonds in hydroxyl groups [24,25]. The peaks at 2927 and 2875 cm$^{-1}$ were assigned to symmetric and asymmetric stretching of –CH$_2$ groups, respectively [26]. The characteristic amide I, II and III bands of the peptide were at approximately 1658, 1531 and 1238 cm$^{-1}$, respectively [27,28].

Compared with adhesive A, no new absorption peaks appeared or disappeared in the FTIR spectrum of adhesive B, indicating that U only denatured the protein but did not react with it. Adhesive C showed a stronger peak at 1160 cm$^{-1}$ than adhesives A and B, assigned to –C–O–C– stretching. However, the absorption peak of adhesive C at 1395 cm$^{-1}$, assigned to COO– bending, was smaller than that of adhesives A and B [29]. In the spectrum of adhesive C, a new peak was observed at 1720 cm$^{-1}$,

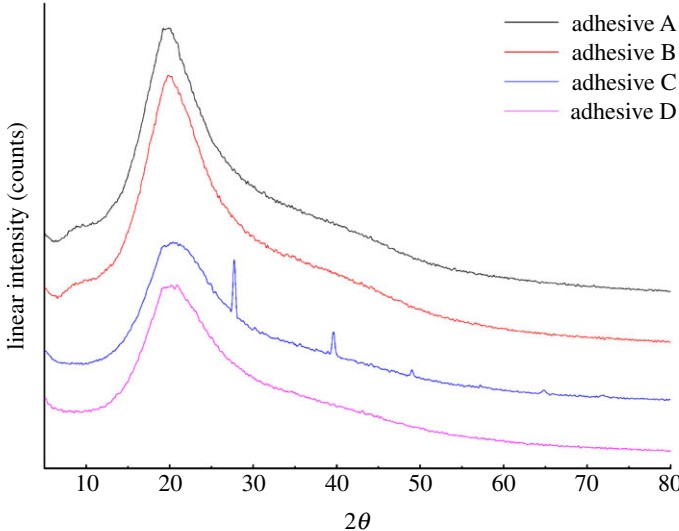

**Figure 5.** XRD patterns of the different adhesives (adhesive A: unmodified PM-based adhesive, adhesive B: PM-based adhesive modified by U, adhesive C: PM-based adhesive modified by U and ECH, adhesive D: PM-based adhesive modified by ECH).

**Table 2.** Crystallinity of the different adhesives (adhesive A: unmodified PM-based adhesive, adhesive B: PM-based adhesive modified by U, adhesive C: PM-based adhesive modified by U and ECH, adhesive D: PM-based adhesive modified by ECH).

| adhesive formulation | adhesive A | adhesive B | adhesive C | adhesive D |
|---|---|---|---|---|
| crystallinity (%) | 2.7 | 2.1 | 11 | 4.4 |

assigned to the stretching of ester C=O bonds [30]. This could arise from esters formed in the reaction of the epoxy groups of ECH with COO– groups of the PM protein, resulting in the formation of new cross-linked network structures, which in turn led to the observed enhanced water resistance [31]. Additionally, the peak at 859 cm$^{-1}$ in the spectrum of adhesive C was attributed to epoxy groups. The high viscosity of adhesive C helped the epoxy group attach to the PM protein and thus aid distribution. Similar findings have been reported in previous studies [30]. Similar to what was found for adhesive C, the 1395 cm$^{-1}$ absorption peak of adhesive D was also weakened relative to that of adhesives A and B, but the intensity of the peak at 1720 cm$^{-1}$ was not as strong as that for adhesive C. This result shows that although the epoxy groups reacted with COO– groups of PM protein, as the latter was not treated with U in the preparation of adhesive D, its reactive groups were less exposed and thus less reactive than those in adhesive C [12].

## 3.5. X-ray diffraction analysis

Figure 5 shows the XRD patterns of the adhesives, and table 2 shows their crystallinities. A broad diffraction peak was observed at $2\theta \approx 20°$ (figure 5), assigned to β-sheet structure [32]. As shown in table 2, the crystallinity decreased from 2.7% for adhesive A to 2.1% for adhesive B because U unfolded and disordered the protein. The crystallinity of adhesive C was 11%, showing that the molecular structure of PM protein in adhesive C became more regular and compact, which was the main reason for the improved water resistance of this adhesive. ECH reacted with the unfolded protein and formed a cross-linked, more crystalline network. Moreover, two new strong peaks were observed at $2\theta = 28°$ and $40°$, which suggested that epoxy groups reacted with the PM protein to form new crystalline phases; this was consistent with the FTIR results. However, although the crystallinity of adhesive D was 4.4%, no new crystal peaks were observed, indicating that no new crystal phases were formed, despite the cross-linking between ECH and PM protein molecules. This reflected the fact that the PM protein was not treated with U: as a consequence, its globular structure was not fully unfolded, which resulted in insufficient cross-linking to form new crystalline zones, so that the crystallinity of adhesive D was lower than that of adhesive C.

**Figure 6.** TG and DTG curves of the different adhesives (adhesive A: unmodified PM-based adhesive, adhesive B: PM-based adhesive modified by U, adhesive C: PM-based adhesive modified by U and ECH, adhesive D: PM-based adhesive modified by ECH).

## 3.6. Thermogravimetry analysis

Figure 6 shows thermogravimetry (TG) and derivative thermogravimetry (DTG) curves of the modified adhesives. The curves can be divided into three main stages: (I) 40–205°C, (II) 205–275°C, and (III) 275–600°C.

In the first stage (I), the weight loss was mainly attributed to evaporation of residual water and decomposition of the tertiary structure of the PM protein. Compared with adhesives A and B, a new peak was observed at 173 and 166°C respectively in the DTG curves of adhesives C and D. This may be owing to the further reaction of ECH with PM protein, resulting in the evolution of steam and gases such as $NH_3$, CO and $CO_2$, resulting in loss of mass [33].

In the second stage (II), the mass loss was mainly owing to the degradation of micromolecules of the PM protein, the decomposition of unstable chemical bonds, and the breaking of intramolecular hydrogen bonds, electrostatic interactions and covalent bonds between amino acid residues [34]. The degradation rate of adhesive B was higher than that of adhesive A, which was explained by the increased disorder in the PM protein caused by the denaturing effect of U.

In the third stage (III), the main weight loss was owing to the decomposition of the protein skeleton [19]. The DTG peak shifted from 307°C for adhesive A to 314°C for adhesive C. This may be owing to the fact that ECH reacted with the PM protein to form many bonds, which strengthened the cross-linked structure of the protein, making it harder to decompose, and thus resulting in higher thermal stability of adhesive C [35]. Therefore, the improved wet shear strength of adhesive C can be explained in terms of changes in thermal stability owing to its cross-linked network.

## 3.7. Scanning electron microscopy analysis

Figure 7 shows SEM images of the surface morphologies [36] of the modified adhesives. For adhesive A, we observed a rough surface with many pores and cracks (figure 7a). These features arose from the lack of cohesive strength in the adhesive during the curing process, in agreement with the findings of Li *et al.* [33]. Moisture can easily enter these pores and destroy the structure of the adhesive, leading to poor water resistance. The addition of U reduced the number of pores and cracks in adhesive B (relative to adhesive A), which partially prevented moisture penetration and improved water resistance. This effect was attributed to U causing the protein molecules to unfold and become more dispersed. The surface of adhesive C was smoother, more uniform [37] and less porous, indicating that ECH formed cross-links with the unfolded PM protein molecules modified by U to form a dense network structure (consistent with the XRD results), increasing the cohesive strength. The dense network structure hindered moisture penetration, which was the main reason for the improvement in water resistance, in agreement with the findings of Salarbashi *et al.* [16]. Compared with adhesives A and B, the surface of adhesive D was smoother and had fewer holes, which might be owing to the reaction between ECH and PM protein to form a network structure. However, without adding U, the globular structure of the protein was not

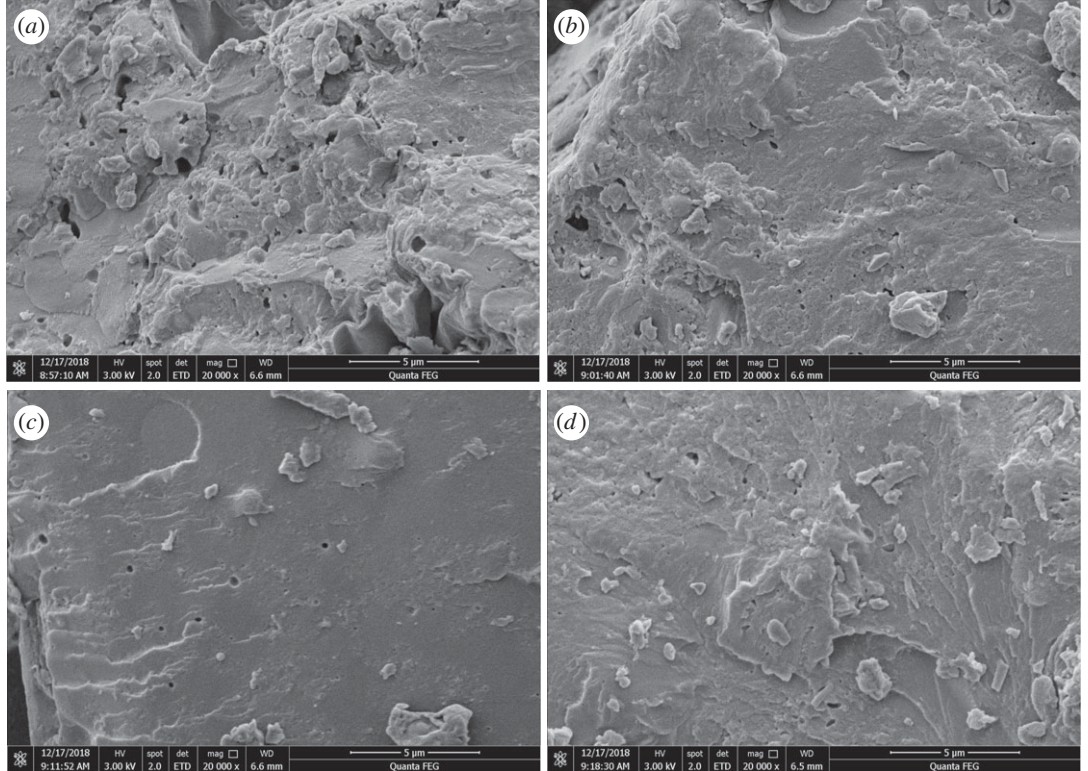

**Figure 7.** SEM images of fracture surfaces of the different adhesives ((a) unmodified PM-based adhesive, (b) PM-based adhesive modified by U, (c) PM-based adhesive modified by U and ECH, (d) PM-based adhesive modified by ECH).

fully unfolded, and its reactive groups were less exposed, resulting in less effective cross-linking between the protein and ECH, so that the surface of adhesive D was not as compact as that of adhesive C.

## 4. Conclusion

A novel PM-based wood adhesive with superior properties was prepared by U and ECH treatment of PM. Using U and ECH obviously improved the dry and wet shear strength of the resulting adhesive to 1.79 MPa and 0.92 MPa respectively, which meets the requirements of the Chinese National Standard. Moreover, the apparent viscosity of the resulting adhesive increased from 819.05 to 24 968 mPa·s and the solid content increased from 21.66% to 25.14%. However, the same enhancement effect could not be achieved by U or ECH treatment of PM alone.

Judging by FTIR results, epoxy groups of ECH reacted with the active groups of the U-treated PM protein and new ester bonds were formed between carboxyl and epoxy groups, promoting the formation of a cross-linked network structure which improved the properties, especially the water resistance. As confirmed by XRD, TG analysis and SEM results, after introducing U and ECH, the resulting adhesive possessed more regular crystalline form, higher thermal stability and smoother protein surface owing to the existence of the cross-linked network structure. All of the above changes in PM protein structure further improved the performance of the resulting adhesive. The present results demonstrated the feasibility of preparing adhesives from PM, expanded the utilization ways of PM and provided a novel approach for deeper use of PM.

Data accessibility. Our data are deposited in the Dryad Digital Repository: https://doi.org/10.5061/dryad.8p3g7p2 [38].
Authors' contributions. All authors contributed to this study. C.C., F.C. and B.L. designed the research study. C.C. performed the assay. C.C. and Y.D. analysed the data. C.C. wrote this paper. C.L., Y.X. and K.L. reviewed the manuscript. All authors gave final approval for publication.
Competing interests. We have no competing interests.
Funding. This work was supported by the National Natural Science Foundation of China (grant no. 21676073), and the 13th Five-year National Key Research and Development Plan (grant no. 2018YFD0401100).
Acknowledgements. We thank Gailing Bao for assistance with FTIR spectroscopy tests and Zheng Zhang for assistance with XRD tests.

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
