## [Reviewer comments · Royal Society Open Science]

Review History

RSOS-191154.R0 (Original submission)

Review form: Reviewer 1

Is the manuscript scientifically sound in its present form?

Yes

Are the interpretations and conclusions justified by the results?

Yes

Is the language acceptable?

Yes

Do you have any ethical concerns with this paper?

Yes

Have you any concerns about statistical analyses in this paper?

No

Recommendation?

Accept with minor revision (please list in comments)

Comments to the Author(s)

The article by Chen et al. describes the employment of peanut meal (PM)-based wood adhesive while the authors propose the enhancement of the adhesive properties by addition of urea (U) and epichlorohydrin (ECH) to the adhesive. Adhesives made with U and ECH showed to possess higher shear strength than without (dry) and exhibited sufficient resistance versus water (sufficient wet shear strength). The authors suggest that the improved properties of adhesive C originate from ECH reaction with functional groups of the PM protein unfolded by U, which results in the generation of three-dimensional network (crosslinking). The language of the article is acceptable for publication, yet some Figures could be improved upon (see below). Furthermore, the addition of a scheme indicating the mechanism of strengthening could make the article more appealing to the readership of Royal Society Open Science.

Therefore, I suggest the publication of this article after addressing the following remarks and minor comments.

Comments:

- The authors state that adhesives C and D have better performance due to “reaction between ECH and PM protein to form a network structure” (i.e. line 219). It may be advisable to visualize the strengthening and water resistance mechanism of the adhesive. Furthermore, it may be beneficial to visualize the assumed reactions (including the reactants such as ECH, U, etc.)?
- Can the authors compare the properties of their adhesive with adhesives on the market and adhesives recently reported in literature?

Minor:

- Please give a reference to the sentence: “China is the world’s principal peanut producer and consumer, with a total output of about 17.7 million tons in 2017.”
- Is it possible to change the color of different bars in the bar chart diagram (Figure 1)? Or at least use clearly different patterns?
- Figure 4: It might be advisable to do a baseline correction for the FTIR spectra.
- The scale bar in Figure 7 is very small, the authors might consider removing the automatic scale bar and adding a custom made (with bigger lettering).

Review form: Reviewer 2

Is the manuscript scientifically sound in its present form?

Yes

Are the interpretations and conclusions justified by the results?

Yes

Is the language acceptable?

Yes

Do you have any ethical concerns with this paper?

No

Have you any concerns about statistical analyses in this paper?

No

Recommendation?

Major revision is needed (please make suggestions in comments)

Comments to the Author(s)

The work is important in the field of science and engineering. The results are interesting. Comments should be considered before the acceptance in the publication process are.:

1. Title should be revised to be short, precise and clear
2. Abstract should be re-written to summarize the work; the abstract should state briefly the purpose of the research, the principal results and major conclusions. An abstract is often presented separately from the article, so it must be able to stand alone the highlights should be more informative and precise
3. In the abstract; "Thermogravimetry tests showed that adhesive C....." add some quantitative data
4. The paper should be clarified in term of uniqueness and advantage of the work compared with other existing works. what is the novelty of this work
5. Why the authors reported on the use of such materials ; rationalize the problem
6. Page section Preparation of PM-based adhesives; what about other tests; add more details
7. Sec Apparent viscosity measurements please add the conditions
8. Page 4 " Preparation of three-ply plywood" This section should be moved to supporting information
9. Under results and discussion; SEM images please deeply discuss the images with respect to porosity and shape see Desalination and Water Treatment 57 (23), 10730-10744; Journal of Water Supply: Research and Technology-Aqua 64 (8), 892-903; Environmental Science and Pollution Research 22 (21), 16721-16731
10. Fourier transform infrared spectroscopy (FTIR); revise the discussion. After "clearly detected at wavenumber of 3323 cm⁻¹ , 2892 cm⁻¹ , 1644 cm⁻¹ , 1422 " add Applied surface science 257 (17), 7746-7751; Journal of cleaner production 172, 2123-2132; Process Safety and Environmental Protection 121, 165-174;
11. discuss the XRD plots, and index please add JCPDS #,
12. The captions of all figures should be with the experimental conditions and parameters;
13. There are many sentences that are long and not clear, Such sentences should be re-written
14. The discussion of the mechanism do not lead to sound conclusions and require revision
15. Discussion could be better impeded with the results with scientific interpretation
16. values in the tables should be rechecked with the correct # sig. figures
17. It is important to show summarize the results clearly in the conclusions; now the conclusion is not suitable
18. Update the introduction with literature as; ISBN13: 9780128047033 Elsevier ; ISBN: 9781522521365 IGI; Colloid and Interface Science Communications 16, 19-24, 2017; Materials Science and Engineering: C 68, 505-511, 2016;
19. After "petroleum resources and the increasing awareness of environmental protection...." Please add Scientific reports 6, 32185; Environmental Science and Pollution Research 22 (21), 16721-16731
20. Graphical abstract should be provided to tell about the work all work in the paper
21. The references should be formatted as per the journal
22. What else is the English, The paper should be checked for typos and grammar errors;
23. Captions of the figure and tables must be with complete information, conditions etc
24. The labels of the y and x axis must be corrected
25. The units in the y and x axis must be presented in a correct form
26. Values in the tables should be of uniform significant figures, please recheck
27. Please improve the conclusion with clear quantitative findings
28. More emphasis on finding and its implication may be mentioned in the conclusion section.
29. Highlights should be prepared as per the format

I WOULD LIKE TO SEE THE REVISED MANUSCRIPT.

Decision letter (RSOS-191154.R0)

02-Sep-2019

Dear Mr Chen:

Title: Properties of peanut meal-based wood adhesives enhanced by urea and epichlorohydrin
Manuscript ID: RSOS-191154

The editor assigned to your manuscript has now received comments from reviewers. We would like you to revise your paper in accordance with the referee and Subject Editor suggestions which can be found below (not including confidential reports to the Editor). Please note this decision does not guarantee eventual acceptance.

Please submit your revised paper before 25-Sep-2019. Please note that the revision deadline will expire at 00.00am on this date. If we do not hear from you within this time then it will be assumed that the paper has been withdrawn. In exceptional circumstances, extensions may be possible if agreed with the Editorial Office in advance. We do not allow multiple rounds of revision so we urge you to make every effort to fully address all of the comments at this stage. If deemed necessary by the Editors, your manuscript will be sent back to one or more of the original reviewers for assessment. If the original reviewers are not available we may invite new reviewers.

Please also include the following statements alongside the other end statements. As we cannot publish your manuscript without these end statements included, if you feel that a given heading is not relevant to your paper, please nevertheless include the heading and explicitly state that it is not relevant to your work.

- Acknowledgements

- Funding statement

Please include a funding section after your main text which lists the source of funding for each author.

RSC Associate Editor:
Comments to the Author:
Please note that we do not require you to cite all references suggested by Reviewer 2 as a condition of publication.

RSC Subject Editor:
Comments to the Author:
(There are no comments.)

Reviewers' Comments to Author:
Reviewer: 1

Comments to the Author(s)

The article by Chen et al. describes the employment of peanut meal (PM)-based wood adhesive while the authors propose the enhancement of the adhesive properties by addition of urea (U) and epichlorohydrin (ECH) to the adhesive. Adhesives made with U and ECH showed to possess higher shear strength than without (dry) and exhibited sufficient resistance versus water (sufficient wet shear strength). The authors suggest that the improved properties of adhesive C originate from ECH reaction with functional groups of the PM protein unfolded by U, which results in the generation of three-dimensional network (crosslinking). The language of the article is acceptable for publication, yet some Figures could be improved upon (see below). Furthermore, the addition of a scheme indicating the mechanism of strengthening could make the article more appealing to the readership of Royal Society Open Science. Therefore, I suggest the publication of this article after addressing the following remarks and minor comments.

Comments:

- The authors state that adhesives C and D have better performance due to "reaction between ECH and PM protein to form a network structure" (i.e. line 219). It may be advisable to visualize the strengthening and water resistance mechanism of the adhesive. Furthermore, it may be beneficial to visualize the assumed reactions (including the reactants such as ECH, U, etc.)?

- Can the authors compare the properties of their adhesive with adhesives on the market and adhesives recently reported in literature?

Minor:

- Please give a reference to the sentence: "China is the world's principal peanut producer and consumer, with a total output of about 17.7 million tons in 2017."
- Is it possible to change the color of different bars in the bar chart diagram (Figure 1)? Or at least use clearly different patterns?
- Figure 4: It might be advisable to do a baseline correction for the FTIR spectra.
- The scale bar in Figure 7 is very small, the authors might consider removing the automatic scale bar and adding a custom made (with bigger lettering).

Reviewer: 2

Comments to the Author(s)

The work is important in the field of science and engineering. The results are interesting. Comments should be considered before the acceptance in the publication process are:

1. Title should be revised to be short, precise and clear
2. Abstract should be re-written to summarize the work; the abstract should state briefly the purpose of the research, the principal results and major conclusions. An abstract is often presented separately from the article, so it must be able to stand alone the highlights should be more informative and precise
3. In the abstract; "Thermogravimetry tests showed that adhesive C....." add some quantitative data
4. The paper should be clarified in term of uniqueness and advantage of the work compared with other existing works. what is the novelty of this work
5. Why the authors reported on the use of such materials ; rationalize the problem
6. Page section Preparation of PM-based adhesives; what about other tests; add more details
7. Sec Apparent viscosity measurements please add the conditions
8. Page 4 " Preparation of three-ply plywood" This section should be moved to supporting information
9. Under results and discussion; SEM images please deeply discuss the images with respect to porosity and shape see Desalination and Water Treatment 57 (23), 10730-10744; Journal of Water Supply: Research and Technology-Aqua 64 (8), 892-903; Environmental Science and Pollution Research 22 (21), 16721-16731
10. Fourier transform infrared spectroscopy (FTIR); revise the discussion. After "clearly detected at wavenumber of 3323 cm⁻¹ , 2892 cm⁻¹ , 1644 cm⁻¹ , 1422 " add Applied surface science 257 (17), 7746-7751; Journal of cleaner production 172, 2123-2132; Process Safety and Environmental Protection 121, 165-174;
11. discuss the XRD plots, and index please add JCPDS #,
12. The captions of all figures should be with the experimental conditions and parameters;
13. There are many sentences that are long and not clear, Such sentences should be re-written
14. The discussion of the mechanism do not lead to sound conclusions and require revision
15. Discussion could be better impeded with the results with scientific interpretation
16. values in the tables should be rechecked with the correct # sig. figures
17. It is important to show summarize the results clearly in the conclusions; now the conclusion is not suitable
18. Update the introduction with literature as; ISBN13: 9780128047033 Elsevier ; ISBN: 9781522521365 IGI; Colloid and Interface Science Communications 16, 19-24, 2017; Materials Science and Engineering: C 68, 505-511, 2016;

19. After “petroleum resources and the increasing awareness of environmental protection...”
Please add Scientific reports 6, 32185; Environmental Science and Pollution Research 22 (21),
16721-16731
 20. Graphical abstract should be provided to tell about the work all work in the paper
 21. The references should be formatted as per the journal
 22. What else is the English, The paper should be checked for typos and grammar errors;
 23. Captions of the figure and tables must be with complete information, conditions etc
 24. The labels of the y and x axis must be corrected
 25. The units in the y and x axis must be presented in a correct form
 26. Values in the tables should be of uniform significant figures, please recheck
 27. Please improve the conclusion with clear quantitative findings
 28. More emphasis on finding and its implication may be mentioned in the conclusion section.
 29. Highlights should be prepared as per the format
- I WOULD LIKE TO SEE THE REVISED MANUSCRIPT.

Author's Response to Decision Letter for (RSOS-191154.R0)

See Appendix A.

RSOS-191154.R1 (Revision)

Review form: Reviewer 1

Is the manuscript scientifically sound in its present form?

Yes

Are the interpretations and conclusions justified by the results?

Yes

Is the language acceptable?

Yes

Do you have any ethical concerns with this paper?

No

Have you any concerns about statistical analyses in this paper?

No

Recommendation?

Accept as is

Comments to the Author(s)

The authors have revised the manuscript to address the comments. I think it is worthy of publishing in its current form.

Review form: Reviewer 2

Is the manuscript scientifically sound in its present form?

Yes

Are the interpretations and conclusions justified by the results?

Yes

Is the language acceptable?

Yes

Do you have any ethical concerns with this paper?

No

Have you any concerns about statistical analyses in this paper?

No

Recommendation?

Accept as is

Comments to the Author(s)

It has been improved

Decision letter (RSOS-191154.R1)

28-Oct-2019

Dear Mr Chen:

Title: Peanut meal-based wood adhesives enhanced by urea and epichlorohydrin

Manuscript ID: RSOS-191154.R1

It is a pleasure to accept your manuscript in its current form for publication in Royal Society Open Science. The chemistry content of Royal Society Open Science is published in collaboration with the Royal Society of Chemistry.

Royal Society of Chemistry
Thomas Graham House
Science Park, Milton Road

Cambridge, CB4 0WF
Royal Society Open Science - Chemistry Editorial Office

RSC Associate Editor:
Comments to the Author:
(There are no comments.)

RSC Subject Editor:
Comments to the Author:
(There are no comments.)

Reviewer(s)' Comments to Author:
Reviewer: 1

Comments to the Author(s)
The authors have revised the manuscript to address the comments. I think it is worthy of publishing in its current form.

Reviewer: 2

Comments to the Author(s)
It has been improved

Appendix A

Manuscript Draft

Manuscript ID: RSOS-191154

Title: Peanut meal-based wood adhesives enhanced by urea and epichlorohydrin

Article Type: Research

Keywords: Peanut meal-based wood adhesive;
Urea;
Epichlorohydrin;
Properties;
Water resistance

Abstract: Peanut meal (PM) has recently emerged as a potential protein source for wood adhesives, due to superior features such as high availability, renewability, and eco-friendliness. However, the poor properties of unmodified PM-based wood adhesives, compared with their petroleum-derived counterparts, limit their use in high-performance applications. In order to promote the application of PM-based wood adhesives in plywood industry, urea (U) and epichlorohydrin (ECH) were used to enhance the properties of the adhesives and the modification mechanism was investigated. PM-based wood adhesives made with U and ECH showed to possess sufficient water resistance and exhibited higher apparent viscosity and solid content than without. FTIR results suggested that U denatured PM protein and expose more reactive groups, allowing ECH reacted better with U-treated PM protein to form a dense, cross-linked network which was the main reason for the improvement of the properties. The crystallinity increased from 2.7% to 11% compared with the control, indicating that the molecular structure of the resultant adhesive modified by U and ECH became more regular and compact due to the cross-linked network structure. TGA tests showed that decomposition temperature of the protein skeleton structure increased from 307°C to 314°C after U and ECH modification. SEM images revealed that using U and ECH for adhesives resulted in a smooth protein surface which prevented moisture penetration and improved water resistance. PM-based adhesives thus represent potential candidates to replace petroleum-derived adhesives in the plywood industry, which will effectively promote the rapid development of eco-friendly adhesives and increase the added value of PM.

<Journal Name>: Royal Society Open Science

<Manuscript ID>: RSOS-191154

<Manuscript Title>: Peanut meal-based wood adhesives enhanced by urea and epichlorohydrin

Dear Editor Dr. Laura Smith:

Thank you very much for your kind information. We highly appreciate the meaningful comments from editor and reviewers concerning our manuscript entitled “Peanut meal-based wood adhesives enhanced by urea and epichlorohydrin”. Those comments are all valuable and very helpful for revising and improving our paper, as well as the important guiding significance to our researches. We have carefully considered the comments and have tried our best to revise the manuscript according to the comments. The amendments are **highlighted in red** in the revised manuscript. Enclosed please find the revised version, which we would like to submit for your kind consideration. Also, a point-by-point list of our response to the reviewers’ and Editor’s comments have been attached.

We would like to express our great appreciation to you and reviewers for comments on our paper. Looking forward to hearing from you.

Thank you and best regards,

Sincerely yours,

Fusheng Chen, Ph.D.

Professor

Henan University of Technology

Zhengzhou, 450001, P.R. China

E-mail: fushengc@haut.edu.cn

Incl.: Responses (in blue) to the comments from editor and reviewers.

Response to the Editor's and reviewers' comments

Editor's comments:

Please also include the following statements alongside the other end statements. As we cannot publish your manuscript without these end statements included, if you feel that a given heading is not relevant to your paper, please nevertheless include the heading and explicitly state that it is not relevant to your work.

- Acknowledgements

- Funding statement

Please include a funding section after your main text which lists the source of funding for each author.

Authors' response: Thanks for editor's careful suggestion. We have added the "Acknowledgements" and the "Funding statement" after the end statements. Please refer to line 240-245 in the revised edition for details.

Reviewer #1:

Comments 1: The authors state that adhesives C and D have better performance due to “reaction between ECH and PM protein to form a network structure” (i.e. line 219). It may be advisable to visualize the strengthening and water resistance mechanism of the adhesive. Furthermore, it may be beneficial to visualize the assumed reactions (including the reactants such as ECH, U, etc.)?

Authors’ response: Thanks for reviewer’s careful comments. We have provided a “Graphical abstract” of this mechanism. Please refer to our Figure 1 to see the assumed reactions.

Figure 1 Graphical abstract

Comments 2: Can the authors compare the properties of their adhesive with adhesives on the market and adhesives recently reported in literature?

Authors’ response: Thanks for reviewer’s kind comments. We have compared the properties of our adhesive with adhesives recently reported in literature. Please refer to Table 1.

Table 1 Properties comparison of the adhesives with different modification treatments

References	Materials	Modification conditions	Properties of the adhesives			
			Solid content (%)	Apparent viscosity (mPa·s)	Dry shear strength (MPa)	Wet shear strength (MPa)
This study	Peanut meal (PM)	Urea (U), Epichlorohydrin (ECH)	25.14	24968	1.79	0.92
Literature 1[1]	Soybean flour	Silane coupling agent	25.90	3500-4000	/	0.98
Literature 2[2]	Soybean flour	Carboxylated styrene-butadiene rubber latex	33.70	125000-150000	0.93	0.97
Literature 3[3]	Sesame protein isolate	Urea, Zinc oxide	25.02	36950	2.00-2.05	1.01

(Note: “/” means that the scholar did not measure the indicator.)

In our study, using U and ECH obviously improved the dry and wet shear strength of the resulting adhesive to 1.79 MPa and 0.92 MPa respectively, which meets the requirements of Chinese National Standard. Moreover, the apparent viscosity of the resulting adhesive increased to 24968 mPa·s and the solid content increased to 25.14% after introducing U and ECH.

Compared with the properties of adhesives in literature 1, there are hardly any differences in solid content and wet shear strength, but the modified adhesive in our work has a higher apparent viscosity than theirs. In general, the operating viscosity limits of plant protein-based adhesives are ranging from 5000 to 25000 mPa·s, depending on the application and the nature of materials to be glued [4].

Compared with the properties of adhesives in literature 2, there are hardly any differences in wet shear strength, but the dry shear strength of our modified adhesive is 92.47% higher than theirs. In addition, the apparent viscosity of their adhesive is too high. If the adhesive is too viscous, the flowability of the adhesive would decrease, and the coating process would become more difficult [5]. The difference in solid content may be due to different addition amounts of protein and modifiers during the preparation of the adhesives.

Compared with the properties of adhesives in literature 3, the wet shear strength of their adhesives is 9.78% higher than ours, but their raw material is sesame protein isolate, which is more expensive and costly. Moreover, the solid content of our adhesive is slightly higher than theirs and the apparent viscosity is more suitable.

1. Question:

Please give a reference to the sentence: “China is the world’s principal peanut producer and consumer, with a total output of about 17.7 million tons in2017.”

Authors’ response: Thanks for reviewer’s careful comments. We have added a reference to the sentence: “China is the world’s principal peanut producer and consumer, with a total output of...” In addition, we revised “17.7” to “17.40” according to the reference. Please refer to line 32 in the revised edition.

2. Question:

Is it possible to change the color of different bars in the bar chart diagram (Figure 1)? Or at least use clearly different patterns?

Authors’ response: Thanks for reviewer’s useful comments. We have changed the color of different bars in the bar chart diagram (Figure 1). Please refer to Figure 1 in the revised edition.

3. Question:

Figure 4: It might be advisable to do a baseline correction for the FTIR spectra.

Authors’ response: Thanks for reviewer’s careful comments. We have done a baseline correction for the FTIR spectra. After baseline correction, the baseline of the FTIR transmittance of each adhesive coincides with the 100% line. However, in order to compare the results of the adhesives, we integrated the FTIR results of the adhesives into one figure, which resulted in that the baseline of some

adhesives did not seem to coincide with the 100% line. A similar representation can be found in Ramesh et al [6]. Please refer to Figure 4 in the revised edition.

4. Question:

The scale bar in Figure 7 is very small, the authors might consider removing the automatic scale bar and adding a custom made (with bigger lettering).

Authors' response: Thanks for reviewer's useful comments. We have revised the scale bar in Figure 7 and make it clearly visible. Please refer to Figure 7 in the revised edition.

Reviewer #2:

1. Question:

Title should be revised to be short, precise and clear.

Authors' response: Thanks for reviewer's insightful suggestion. We have revised the title to "Peanut meal-based wood adhesives enhanced by urea and epichlorohydrin". Please refer to line 1 in the revised edition.

2. Question:

Abstract should be re-written to summarize the work; the abstract should state briefly the purpose of the research, the principal results and major conclusions. An abstract is often presented separately from the article, so it must be able to stand alone the highlights should be more informative and precise.

Authors' response: We appreciate the reviewer very much for the very constructive comments. We have rewritten the abstract, containing the purpose of the research, the principal results and major conclusions. Please refer to line 3-18 in the revised edition.

3. Question:

In the abstract; "Thermogravimetry tests showed that adhesive C....." add some quantitative data.

Authors' response: Thanks for reviewer's useful comments. We have added some quantitative data after "Thermogravimetry tests showed that adhesive C...." in the abstract. Please refer to line 14 in the revised edition.

4. Question:

The paper should be clarified in term of uniqueness and advantage of the work compared with other existing works. What is the novelty of this work.

Authors' response: Thanks for reviewer's constructive comments.

Compared with other existing works, the uniqueness of our work is as follows.

(1). This research proved the feasibility of preparing adhesives from peanut meal (PM), expanded the utilization ways of PM, and provided a novel approach for deeper utilization of PM.

(2). The material we used was PM, which was rarely applied in wood adhesive

industry before.

(3). Urea (U) and epichlorohydrin (ECH) were used to improve the properties of PM-based wood adhesives for the first time.

(4). The mechanism of improving the performance of PM-based wood adhesives by U and ECH was first studied.

Compared with other existing works, the advantages of our work are as follows.

(1). Compared with petroleum-based adhesives, the raw material of our work is mainly PM, which is a cheap and renewable protein resource. In addition, the wet shear strength of our resulting adhesive also meets the requirements of Chinese National Standard.

(2). Compared with other plant protein-based adhesives, our adhesives have better comprehensive properties. Please refer to Table 1.

Firstly, compared with the properties of adhesives in literature 1, there are hardly any differences in solid content and wet shear strength, but the modified adhesive in our work has a higher apparent viscosity than theirs. In general, the operating viscosity limits of plant protein-based adhesives are ranging from 5000 to 25000 mPa·s, depending on the application and the nature of materials to be glued [4].

Secondly, compared with the properties of adhesives in literature 2, there are hardly any differences in wet shear strength, but the dry shear strength of our modified adhesive is 92.47% higher than theirs. In addition, their adhesives are too viscous. If the viscosity is too large, the flowability of the adhesive would decrease, and the coating process would become more difficult [5]. The difference in solid content may be due to the difference in protein and modifiers addition amount during the preparation of the adhesives.

Thirdly, compared with the properties of adhesives in literature 3, the wet shear strength of their adhesives is 9.78% higher than ours, but their raw material is sesame protein isolate, which is more expensive and costly. Moreover, the solid content of our adhesive is slightly higher than theirs and the apparent viscosity is more suitable.

Table 1 Properties comparison of the adhesives with different modification treatments

References	Materials	Modification conditions	Properties of the adhesives			
			Solid content (%)	Apparent viscosity (mPa·s)	Dry shear strength (MPa)	Wet shear strength (MPa)
This study	Peanut meal (PM)	Urea (U), Epichlorohydrin (ECH)	25.14	24968	1.79	0.92
Literature 1[1]	Soybean flour	Silane coupling agent	25.90	3500-4000	/	0.98
Literature 2[2]	Soybean flour	Carboxylated styrene-butadiene rubber latex	33.70	125000-150000	0.93	0.97
Literature 3[3]	Sesame protein isolate	Urea, Zinc oxide	25.02	36950	2.00-2.05	1.01

(Note: “/” means that the scholar did not measure the indicator.)

The novelty of this work is as follows.

- (1). This paper provides a new and effective method to prepare wood adhesives using PM as the raw material.
- (2). U and ECH were used to improve the properties of PM-based wood adhesives for the first time.
- (3). The mechanism of improving the properties of PM-based wood adhesives by U and ECH was first studied.
- (4). Using U and ECH obviously improved the wet shear strength of the resulting adhesive to 0.92 MPa, which meets the requirements of Chinese National Standard.
- (5). The cross-linked network formed by ECH and U-treated PM is the main reason for performance improvement.

5. Question:

Why the authors reported on the use of such materials; rationalize the problem.

Authors' response: Thanks for reviewer's useful comments. The reasons we choose PM are as follows.

With the rapid development of the artificial plywood industry, the production and use of wood adhesives have steadily increased in recent years. At present, the vast majority of the adhesives used in plywood composites are based on formaldehyde. Typical examples are U-, phenol-, and melamine-formaldehyde resins. Because of their excellent bonding properties, high water resistance, and low cost, these are very important in the wood adhesive industry [7, 8]. However, the release of formaldehyde (derived from non-renewable petroleum resources) during the production and use of these resins is a crucial problem that needs to be addressed. The depletion of petroleum resources and the increasing awareness of environmental protection issues [9, 10] have created demand for adhesives that are free of formaldehyde and made from renewable bioresources [11].

Plant proteins are high-quality renewable resources for formaldehyde-free wood adhesives, and have received widespread attention. A number of previous studies have focused on plant protein-based adhesives, including adhesives based on soy [12], cotton [13], wheat [14], and sesame proteins [3]. Nevertheless, there are few studies on the preparation of wood adhesives based on peanut proteins. China is the world's principal peanut producer and consumer, with a total output of about 17.40 million tons in 2017 [15]. These abundant resources provide sufficient raw materials for the study and development of PM protein and processed products. As a main byproduct of peanut oil processing, PM is primarily used as animal feed or crop fertilizer due to its high protein content, which results in severe waste of resources. Therefore, the efficient use of PM represents an issue of significant interest. Three major components, arachin, conarachin I, and conarachin II, containing a large number of reactive groups, account for >75% of the total protein content in peanut. Chemical reactions can take place between these reactive groups and modifiers, which makes it possible to prepare wood adhesives from PM [16].

Moreover, using PM as raw material to prepare adhesives can not only

provide an alternative to petroleum-based adhesives, but also promote the efficient and high-quality utilization of PM. If the PM can be prepared into adhesives and applied in plywood industry, it will produce more economic benefits than before when the PM is made into animal feed or crop fertilizer.

The reasons we choose U and ECH are as follows.

The poor properties of unmodified PM-based wood adhesives limit their use in high-performance applications. Methods that can modify plant protein-based adhesives to enhance their properties include denaturation [17] or modifications with cross-linking agents [1], synthetic resins [18], nanomaterials [19, 20], and biomimetic materials [21, 22]. These modifications have produced effective improvements in the properties of plant protein-based adhesives [23].

Considering the high content of reactive groups in peanut protein, chemical modifications were selected to enhance the properties of PM-based adhesives. ECH can react with the reactive groups of the proteins via its epoxy group and generate new firm chemical bonds [24], which make the interior of the protein structure more stable and less susceptible to damage from the external environment. However, these reactive groups are generally embedded inside the protein and cannot easily interact with the ECH, which makes it difficult to meet the needs of ECH modification alone. Therefore, it is necessary to add a denaturing agent before adding ECH in order to better react. U is used as a denaturing agent to break the structure of the protein and expose the functional groups [25], so that they can react with ECH sufficiently, thus further improving the properties of PM-based adhesives. Therefore, U and ECH were selected improve the properties of PM-based adhesives.

Modifications to the Introduction are shown in line 28-32, 37-38, 44-52 in the revised edition.

6. Question:

Page section Preparation of PM-based adhesives; what about other tests; add more details.

Authors' response: Thanks for reviewer's precise comments. We have added more details in the section "Preparation of PM-based adhesives". As for the FTIR, XRD, TGA and SEM tests, the preparation of the modified adhesives is consistent with this section. Please refer to line 64-72 in the revised edition.

7. Question:

Sec Apparent viscosity measurements please add the conditions.

Authors' response: Thanks for reviewer's kind suggestion. We have added the conditions in the section "Apparent viscosity measurements". The apparent viscosity was determined based on the method described by Wei et al. [3] with some modifications. Please refer to line 82-85 in the revised edition.

8. Question:

Page 4 “Preparation of three-ply plywood”. This section should be moved to supporting information.

Authors’ response: Thanks for reviewer’s careful comments. We have moved “Preparation of three-ply plywood” to supporting information.

9. Question:

Under results and discussion; SEM images please deeply discuss the images with respect to porosity and shape see Desalination and Water Treatment 57 (23), 10730-10744; Journal of Water Supply: Research and Technology-Aqua 64 (8), 892-903; Environmental Science and Pollution Research 22 (21), 16721-16731.

Authors’ response: Thanks for reviewer’s precise comments. We have updated the discussion of SEM images with the references recommended by the reviewer. Please refer to line 208, 214 in the revised edition.

It is not appropriate to use the term “porosity” because we are meant to express “hole” rather than “porosity”. We are so sorry for the ambiguity aroused by negligence. Then we changed the phrase “lower porosity” to “fewer holes” in the following texts. Please refer to line 219 in the revised edition.

10. Question:

Fourier transform infrared spectroscopy (FTIR); revise the discussion. After “clearly detected at wavenumber of 3323 cm⁻¹, 2892 cm⁻¹, 1644 cm⁻¹, 1422 ” add Applied surface science 257 (17), 7746-7751; Journal of cleaner production 172, 2123-2132; Process Safety and Environmental Protection 121, 165-174;

Authors’ response: Thanks for reviewer’s useful comments. We have revised the discussion about FTIR analysis. In addition, the references recommended by the reviewer were cited. Please see refer to line 157-158 in the revised edition.

11. Question:

discuss the XRD plots, and index please add JCPDS #.

Authors’ response: Thanks for reviewer’s careful comments. The JCPDS card is the basis for X-ray qualitative phase analysis. The X-ray qualitative phase analysis is a method of comparing the measured diffraction pattern of the unknown phase with the standard data of the known crystal structure phase in the JCPDS card.

In our work, the XRD tests were measured using a D8 ADVANCE diffractometer (Bruker, Germany). We compared our XRD results with the data in the International Center for Diffraction Data, but we did not find the corresponding JCPDS#. It may be because PM, the main raw material of the adhesive, is amorphous. The two new diffraction peaks appearing in adhesive C indicate that due to the presence of the cross-linked network structure, the protein in adhesive C has a new crystal form, rather than a new crystalline substance. So we did not find the corresponding JCPDS#.

12. Question:

The captions of all figures should be with the experimental conditions and

parameters;

Authors' response: Thanks for reviewer's careful comments. We have revised the captions of all figures and made them consistent with the experimental conditions and parameters. Please refer to line 361-381 in the revised edition.

13. Question:

There are many sentences that are long and not clear, Such sentences should be re-written.

Authors' response: Thanks for reviewer's careful comments. We scrutinized the language of the whole manuscript and rewrote the sentences that are long and not clear. In addition, we invited professional language polishing company (ELIXIGEN) to review and revise this paper again.

14. Question:

The discussion of the mechanism does not lead to sound conclusions and require revision.

Authors' response: Thanks for reviewer's insightful suggestion. In order to elucidate the modification mechanism of U and ECH, the FTIR, XRD, TGA and SEM tests were carried out. We have revised the conclusions. Please refer to line 224-237 in the revised edition.

15. Question:

Discussion could be better impended with the results with scientific interpretation.

Authors' response: Thanks for the reviewer's useful comments. We have revised some discussions with scientific interpretation according to the references recommended by the reviewer. Please refer to line 157-158, 208 and 219 in the revised edition.

16. Question:

Values in the tables should be rechecked with the correct # sig. figures.

Authors' response: Thanks for reviewer's careful comments. We have rechecked values in the tables and revised them to be of uniform significant figures. Please refer to Table 1, 2 and line 12, 131-132, 178 in the revised edition for details.

17. Question:

It is important to show summarize the results clearly in the conclusions; now the conclusion is not suitable.

Authors' response: Thanks for reviewer's insightful suggestion. Previous conclusion was not suitable enough. We have rewritten the conclusions. Please refer to line 224-237 in the revised edition for details.

18. Question:

Update the introduction with literature as; ISBN13: 9780128047033 Elsevier; ISBN: 9781522521365 IGI; Colloid and Interface Science Communications 16, 19-24, 2017; Materials Science and Engineering: C 68, 505-511, 2016;

Authors' response: Thanks for reviewer's useful comments. We have updated the references recommended by the reviewer. Please refer to line 41-42 in the revised edition for details.

19. Question:

After “petroleum resources and the increasing awareness of environmental protection....” Please add Scientific reports 6, 32185; Environmental Science and Pollution Research 22 (21), 16721-16731

Authors' response: Thanks for reviewer's useful comments. The references recommended by the reviewer were added after “petroleum resources and the increasing awareness of environmental protection....” Please refer to line 26 in the revised edition for details.

20. Question:

Graphical abstract should be provided to tell about the work all work in the paper

Authors' response: Thanks for reviewer's useful suggestion. A graphical abstract for all work in the paper is as follows.

Figure 1 Graphical abstract

21. Question:

The references should be formatted as per the journal.

Authors' response: Thanks for reviewer's careful comments. The references have been formatted according to the *Author guidelines* of RSOS. Please refer to line 247-352 in the revised edition for details.

22. Question:

What else is the English, The paper should be checked for typos and grammar errors.

Authors' response: Thanks for reviewer's careful comments. We have checked for typos and grammar errors. In addition, we invited professional language polishing company (ELIXIGEN) to review and revise this paper again.

23. Question:

Captions of the figure and tables must be with complete information, conditions etc.

Authors' response: Thanks for reviewer's useful comments. We have revised the captions of all figures and tables with the experimental conditions and parameters. Please refer to 355-385 in the revised edition for details.

24. Question:

The labels of the y and x axis must be corrected.

Authors' response: Thanks for reviewer's careful comments. In Figure 2, there was a missing y-axis label of "Apparent viscosity". We have revised it in Figure 2. In addition, we have rechecked other figures. Please refer to Figures 1 to 7 in the revised edition for details.

25. Question:

The units in the y and x axis must be presented in a correct form

Authors' response: Thanks for reviewer's useful comments. We have rechecked units of the axes of all the figures. Please refer to Figures 1 to 7 in the revised edition for details.

26. Question:

Values in the tables should be of uniform significant figures, please recheck.

Authors' response: Thanks for reviewer's careful comments. We have rechecked values in the tables and revised them to be of uniform significant figures. Please refer to Table 1, 2 and line 12, 131-132, 178 in the revised edition for details.

27. Question:

Please improve the conclusion with clear quantitative findings

Authors' response: Thanks for reviewer's insightful suggestion. We have rewritten the conclusions with clear quantitative findings. Please refer to line 224-229 in the revised edition for details.

28. Question:

More emphasis on finding and its implication may be mentioned in the conclusion section.

Authors' response: Thanks for reviewer's useful comments. More emphasis on finding and its implication has been mentioned in the conclusion section. Please refer to line 224-237 in the revised edition for details.

29. Question:

Highlights should be prepared as per the format.

Authors' response: Thanks for reviewer's insightful suggestion. The highlights have been prepared as per the format.

Highlights

- * Urea (U) and epichlorohydrin (ECH) affect the water resistance of peanut meal adhesive.
- * The wet shear strength of the U/ECH-modified adhesive (C) meets interior use requirements.
- * The superior properties of adhesive C are due to its cross-linked network structure.
- * Peanut meal adhesives are suitable eco-friendly alternatives to petroleum-based ones.

References

- 1 Li C, Li H, Zhang S, Li J. 2014 Preparation of Reinforced Soy Protein Adhesive Using Silane Coupling Agent as an Enhancer. *Bioresources*. **9**, 5448-5460. (DOI: 10.15376/biores.9.3.5448-5460)
- 2 Li H, Kang H, Wei Z, Zhang S, Li J. 2016 Physicochemical properties of modified soybean-flour adhesives enhanced by carboxylated styrene-butadiene rubber latex. *Int. J. Adhes. Adhes.* **66**, 59-64. (DOI: 10.1016/j.ijadhadh.2015.12.008)
- 3 Wei X, Wang X, Li Y, Ma Y. 2017 Properties of a new renewable sesame protein adhesive modified by urea in the absence and presence of zinc oxide. *Rsc Adv.* **7**, 46388-46394.(DOI: 10.1039/C7RA07578B)
- 4 Li J, Luo J, Li X, Zhao Y, Gao Q, Li J. 2015 Soybean meal-based wood adhesive enhanced by ethylene glycol diglycidyl ether and diethylenetriamine. *Ind. Crops Prod.* **74**, 613-618. (DOI: 10.1016/j.indcrop.2015.05.066)
- 5 Li H, Li C, Gao Q, Zhang S, Li J. 2014 Properties of soybean-flour-based adhesives enhanced by attapulgite and glycerol polyglycidyl ether. *Ind. Crops Prod.* **59**, 35-40. (DOI: 10.1016/j.indcrop.2014.04.041)
- 6 Ramesh, K., Melzner, F., Griffith, A. W., Gobler, C. J., Rouger, C., Tasdemir, D., Nehrke, G. 2018 In vivo characterization of bivalve larval shells: a confocal Raman microscopy study. *J. R. Soc. Int.* **15**, 20170723. (DOI: 10.1098/rsif.2017.0723)
- 7 Luo J, Luo J, Gao Q, Li J. 2015 Effects of heat treatment on wet shear strength of plywood bonded with soybean meal-based adhesive. *Ind. Crops Prod.* **63**, 281-286.(DOI: 10.1016/j.indcrop.2014.09.054)
- 8 Vnučec D, Goršek A, Kutnar A, Mikuljan M. 2015 Thermal modification of soy proteins in the vacuum chamber and wood adhesion. *Wood Sci. Technol.* **49**, 225-239. (DOI: 10.1007/s00226-014-0685-5)
- 9 Saleh TA, Al-Shalalfeh MM, Al-Saadi AA. 2016 Graphene Dendrimer-stabilized silver nanoparticles for detection of methimazole using Surface-enhanced Raman scattering with computational assignment. *Sci. Rep.* **6**, 32185. (DOI: 10.1038/srep32185)
- 10 Saleh, T. A. 2015 Isotherm, kinetic, and thermodynamic studies on Hg(II) adsorption from aqueous solution by silica- multiwall carbon nanotubes. *Environmental Science & Pollution Research International.* **22**, 16721-16731.
- 11 Zhang, B., Fan, B., Huo, P., Gao, Z.-H. 2017 Improvement of the water resistance of soybean protein-based wood adhesive by a thermo-chemical treatment approach. *International Journal of Adhesion and Adhesives.* **78**, 222-226. (<https://doi.org/10.1016/j.ijadhadh.2017.08.002>)

- 12 Lei H, Du G, Wu Z, Xi X, Dong Z. 2014 Cross-linked soy-based wood adhesives for plywood. *Int. J. Adhes. Adhes.* **50**, 199-203. (DOI: 10.1016/j.ijadhadh.2014.01.026)
- 13 Cheng HN, Dowd MK, He Z. 2013 Investigation of modified cottonseed protein adhesives for wood composites. *Ind. Crops Prod.* **46**, 399-403. (DOI: 10.1016/j.indcrop.2013.02.021)
- 14 Nordqvist P, Khabbaz F, Malmström E. 2010 Comparing bond strength and water resistance of alkali-modified soy protein isolate and wheat gluten adhesives. *Int. J. Adhes. Adhes.* **30**, 72-79. (DOI: 10.1016/j.ijadhadh.2009.09.002)
- 15 Gong A, Shi A, Liu H, Yu H, Liu, L, Lin, W, Wang Q. 2018 Relationship of chemical properties of different peanut varieties to peanut butter storage stability. *J. Integ Agric.* **17**, 1003-1010. (DOI: 10.1016/S2095-3119(18)61919-7)
- 16 Yan YS, Lin XD, Zhang YS, Lei W, Wu K, Huang SZ. 2005 Isolation of peanut genes encoding arachins and conglutins by expressed sequence tags. *Plant Sci.* **169**, 439-445. (DOI: 10.1016/j.plantsci.2005.04.010)
- 17 Luo J, Li X, Zhang H, Gao Q, Li J. 2016 Properties of a soybean meal-based plywood adhesive modified by a commercial epoxy resin. *Int. J. Adhes. Adhes.* **71**, 99-104. (DOI: 10.1016/j.ijadhadh.2016.09.002)
- 18 Gui C, Wang G, Di W, Jin Z, Liu X. 2013 Synthesis of a bio-based polyamidoamine-epichlorohydrin resin and its application for soy-based adhesives. *Int. J. Adhes. Adhes.* **44**, 237-242. (DOI: 10.1016/j.ijadhadh.2013.03.011)
- 19 Alswat AA, Ahmad MB, Saleh TA. 2017 Preparation and Characterization of Zeolite/Zinc Oxide-Copper Oxide Nanocomposite: Antibacterial Activities. *J. Colloid. Interface. Sci.* **16**, 19-24. (DOI: 10.1016/j.colcom.2016.12.003)
- 20 Salarbashi D, Mortazavi SA, Noghabi MS, Bazzaz BSF, Sedaghat N, Ramezani M, Shahabi-Ghahfarrokhi I. 2016 Development of new active packaging film made from a soluble soybean polysaccharide incorporating ZnO nanoparticles. *Carbohydr. Polym.* **140**, 220-227. (DOI: 10.1016/j.carbpol.2015.12.043)
- 21 Alswat AA, Ahmad MB, Saleh TA, Hussein MZB, Ibrahim NA. 2016 Effect of zinc oxide amounts on the properties and antibacterial activities of zeolite/zinc oxide nanocomposite. *Mater. Sci. Eng. C* **68**, 505-511. (DOI: 10.1016/j.msec.2016.06.028)
- 22 Zhang YH, Zhu WQ, Gao ZH, Gu JY. 2015 Effects of crosslinking on the mechanical properties and biodegradability of soybean protein-based composites. *J. Appl. Polym. Sci.* **132**, 41387. (DOI: 10.1002/app.41387)
- 23 Yuan C, Chen M, Luo J, Li X, Gao Q, Li J. 2017 A novel water-based process produces eco-friendly bio-adhesive made from green cross-linked soybean soluble polysaccharide and soy protein. *Carbohydr. Polym.* **169**, 417-425. (DOI: 10.1016/j.carbpol.2017.04.058)
- 24 Mousavi SY, Huang J, Li K. 2018 Investigation of poly (glycidyl methacrylate-co-styrene) as a curing agent for soy-based wood adhesives. *Int. J. Adhes. Adhes.* **82**, 67-71. (DOI: 10.1016/j.ijadhadh.2017.12.017)
- 25 Zhang Z. 2007 Urea-Modified Soy Globulin Proteins (7S and 11S): Effect of Wettability and Secondary Structure on Adhesion. *J. Am. Oil Chem. Soc.* **84**, 853-857. (DOI: 10.1007/s11746-007-1108-7)